# Protein–Protein Connections—Oligomer, Amyloid and Protein Complex—By Wide Line ^1^H NMR

**DOI:** 10.3390/biom11050757

**Published:** 2021-05-18

**Authors:** Mónika Bokor, Ágnes Tantos

**Affiliations:** 1Wigner Research Centre for Physics, Institute for Solid State Physics and Optics, 1121 Budapest, Hungary; 2Research Centre for Natural Sciences, Institute of Enzymology, 1117 Budapest, Hungary; tantos.agnes@ttk.hu

**Keywords:** α-helix, amyloid, β-sheet, hydration, intrinsically disordered proteins, oligomer, protein–protein interactions, wide-line ^1^H NMR

## Abstract

The amount of bonds between constituting parts of a protein aggregate were determined in wild type (WT) and A53T α-synuclein (αS) oligomers, amyloids and in the complex of thymosin-β_4_–cytoplasmic domain of stabilin-2 (Tβ_4_-stabilin CTD). A53T αS aggregates have more extensive βsheet contents reflected by constant regions at low potential barriers in difference (to monomers) melting diagrams (*MD*s). Energies of the intermolecular interactions and of secondary structures bonds, formed during polymerization, fall into the 5.41 kJ mol^−1^ ≤ *E*_a_ ≤ 5.77 kJ mol^−1^ range for αS aggregates. Monomers lose more mobile hydration water while forming amyloids than oligomers. Part of the strong mobile hydration water–protein bonds break off and these bonding sites of the protein form intermolecular bonds in the aggregates. The new bonds connect the constituting proteins into aggregates. Amyloid–oligomer difference *MD* showed an overall more homogeneous solvent accessible surface of A53T αS amyloids. From the comparison of the nominal sum of the *MD*s of the constituting proteins to the measured *MD* of the Tβ_4_-stabilin CTD complex, the number of intermolecular bonds connecting constituent proteins into complex is 20(1) H_2_O/complex. The energies of these bonds are in the 5.40(3) kJ mol^−1^ ≤ *E*_a_ ≤ 5.70(5) kJ mol^−1^ range.

## 1. Introduction

One of the main reasons intrinsically disordered proteins (IDPs) are so important in different physiological processes is that they are capable of forming a bewildering variety of interactions. They can undergo induced folding or unfolding, they are able to form physiological or pathological aggregates, and even form fuzzy complexes, where disorder is retained during the interaction [1]. Describing the molecular details of the different interaction types of IPDs is crucial for the clear understanding their mechanisms of function and for the interference with the pathological processes caused by their erroneous behavior.

In our work presented here, two IDP interaction systems were studied to gain information about the bonds holding the protein associations—oligomer, amyloid, and complex—together. One system contains wild type and mutant α-synuclein (αS) in the forms of oligomers and amyloids. Detailed structural and preparation information can be found in [2,3]. The other system consists of two proteins—thymosin-β_4_ (Tβ_4_) and cytoplasmic domain of stabilin-2 (stabilin CTD)—which form a 1:1 complex (see [4,5] for structural information and for details of preparation).

αS is a 140-amino-acid protein mainly located at presynaptic terminals and is abundant in the brain [6]. The exact physiological role of αS remains unclear, but it’s thought to be involved in the regulation of synaptic vesicle mobility [7] through binding to VAMP [8]. Pathological aggregation of αS is the leading cause of Parkinson’s disease (PD), and some mutations of αS, such as A53T, cause familial occurrences of the disease [9,10]. A53T mutation results in faster αS accumulation [11]. When the balance between the production and clearance of αS is disturbed, the monomeric αS aggregates and misfolds into oligomers, then amyloid fibrils and finally Lewy bodies [12]. The exact pathogenesis of PD is still unknown, but more and more evidence suggests that the oligomeric form of αS plays a vital part in the pathogenesis of PD [12,13,14,15,16]. Historically, αS has been considered as an IDP in the healthy, physiological state [17,18]. IDPs exist as dynamic protein ensembles with fluxional structures under physiological conditions that can adopt different conformations depending on their external stimuli, thus enabling the complex signaling and regulatory requirements of higher biological systems [19,20].

The structural traits that lie behind the stronger amyloidogenic propensity of the disease-related mutants are intensively studied and have revealed many important features. The helical content of wild type (WT) αS is minimally affected by the A53T mutation, while there is an increase in the β-sheet content of the A53T mutant-type in comparison to the WT protein [18,21,22,23,24]. Long-range intramolecular interactions between the different regions either are less abundant or disappear upon A53T mutation [25,26,27], indicating that the NAC (non-AB component) region is more solvent-exposed upon A53T mutation [28]. As this segment is indispensable for the amyloid formation, its easier accessibility explains the increased aggregation propensity of the A53T mutant [22,23,26,29,30,31,32,33]. The structures of the A53T αS are thermodynamically more preferred than the structures of the WT αS [26]. 

Our understanding of amyloid structure lately has been improved a lot by innovations in cryo-electron microscopy, electron diffraction and solid-state NMR. The results show expected cross-β amyloid structure and an unexpectedly diverse and complex amyloid fold [34]. The amyloid cores are constituted of extensive mainchain hydrogen bonding between adjacent β-strands within the stacked layers, and close interdigitation of sidechains. The amyloid fibrils are dynamic, with monomers and/or oligomers dissociating from their ends [35,36].

Isolated Tβ_4_, stabilin CTD and their 1:1 complex create a system where the bonds formed in the complex can be studied. 

Tβ_4_ is an IDP with moonlighting functions. It can sequester actin [37] and it has important role in the regulation of the formation and modulation of the actin cytoskeleton [38]. Many different functions were attributed to Tβ_4_. Such functions are, e.g., endothelial cell differentiation [39], angiogenesis [40] and wound repair [41]. Tβ_4_ can increase the concentration of metalloproteinases, which is important in cell migration [41], the underlying process of which is still unclear. The multiple functions of Tβ_4_ are probably related to that, it is an IDP and has very little transient secondary structure in solution phase [42]. It binds often weakly to its physiological partners, and forms structurally heterogeneous complexes with them [43], but it can fold upon binding.

Stabilin-2 is an endocytic receptor for hyaluronic acid, and its C-terminal domain (CTD) binds Tβ_4_ [4,44]. The observation that Tβ_4_ colocalizes with stabilin-2 in the phagocytic cup suggests that the complex of Tβ_4_ with stabilin-2 CTD is involved in the phagocytosis of apoptotic cells [45]. It was also shown that knockdown or overexpression of Tβ_4_ decreased and enhanced the phagocytic activity of stabilin-2, respectively. While the exact molecular mechanism of this is yet unclear, it was found that Tβ_4_ engages in multiple fuzzy interactions with stabilin-2 CTD. It is capable of mediating its distinct yet specific interactions without adapting distinct folded structures in the different complexes [42]. Weak binding was confirmed between Tβ_4_ and stabilin-2 CTD and it was suggested that the proteins become slightly more disordered in the complex [4,5].

The purpose of this experimental study is to determine the amount and quality of bonds between constituting parts of a protein aggregate in wild type (WT) and A53T α-synuclein (αS) oligomers, amyloids and in the complex of thymosin-β_4_–cytoplasmic domain of stabilin-2. The selection of the studied proteins represents a system in which the constituents are different in their degree of polymerization and another system where the constituents interact with each other. This work presents as novelty compared to the previous works [2,3,4,5] the differences between α-Ss of different degrees of polymerization and sheds light on the bonds between the constituting proteins in the Tβ_4_-stabilin CTD complex.

## 2. Materials and Methods

The studied proteins were prepared, and their qualities were checked as in [2,3,4,5]. Briefly, untagged versions of α-synuclein variants were expressed in *E. coli* cells and purified using anion exchange chromatography, while stabilin CTD was expressed with an N-terminal His-tag and purified on a Ni-NTA affinity column. The wide-line ^1^H NMR signals (see Appendix A and Supporting Information of [3]) on the time scale are composed of more components of different origins (ice, protein, mobile water) and time constants. The intensities of the components with the longest decaying-time constants give the amounts of the mobile hydration water. Melting diagrams (*MD*s) are the relative amounts of the proteins’ mobile water measured by intensities of wide-line ^1^H NMR signals vs. normalized functional temperature. The mobile hydration differences were calculated from *MD*s of WT and A53T αS monomers, oligomers and amyloids in [3], and from *MD*s of Tβ_4_, stabilin-2 CTD and their 1:1 complex in [4,5]. In this work, potential barriers corresponding to normalized functional temperature are used. Potential barriers are calculated from functional (or absolute) temperature by first normalizing it with the melting point of water (273.15 K) and then scaling this normalized temperature with the melting heat of ice (*Q* = 6.01 kJ mol^−1^), *E*_a_ = *T*·*Q*/*T*_m_, where *T* is the absolute temperature and *T*_m_ is the melting point of water. The relative amount of mobile water is converted to molar amount of mobile water per amino acid residues, *naa*. The *MD*s are given as *naa* vs. *E*_a_. The motion of water is controlled by the potential barrier.

## 3. Results

### 3.1. The α-Synuclein System

Mobile hydration differences (monomer *MD* subtracted from oligomer (o-m) and amyloid (a-m) *MD*s) were calculated for αS variants of different polymerization degrees. The difference *MD*s for the o-m and the *a-m* cases are very similar for each variant (Figure 1).

The derivative ratios are depicted in the insets of the figures and they reflect very sensitively the changes in the trends of the melting curves. However, they have characteristically different shapes for the WT and the A53T variants. While the WT αS o-m difference *MD* has two sections with distinct slopes, A53T αS has three. The additional low-potential-barrier (*E*_a_) sections in A53T difference curves can be attributed to the excess β-sheet contents [22,23,24,25] of the mutant compared to WT. The additional β-sheet content is reflected by the low *E*_a_ constant *naa* (mol H_2_O per mol amino acid residue) or only little raising *naa* sections in o-m and a-m differences, respectively (Figure 1b). This is supported by the inverse preferences to form helices for alanine and to support β-sheet structures for threonine [46], i.e., A53T αS is more prone to have β-sheets than WT αS. This suggestion is affirmed by the comparison of o-m and a-m differences for the WT and the A53T αS variants (Figure 2).

The WT and the A53T o-m differences are equal in the potential barrier region 5.41 kJ mol^−1^ ≤ *E*_a_ ≤ 5.77 kJ mol^−1^ (Figure 2a). This region belongs to interactions incorporating the monomers into oligomers or amyloids and to the secondary structures forming in the process of polymerization. The loss of mobile hydration water is the most intensive at potential barrier values near to the melting point of bulk water (6.01 kJ mol^−1^). The slope of the decrease is less steep in the case of o-m curves than for o-a curves (Figure 1 and Figure 2). In this region, close to the melting point of bulk water, the most strongly bound water molecules start to move. The mobile hydration water loss is the greatest for the a-m curves, here is where we can detect the largest difference compared to the monomers. The monomers lose much more hydration water, which is visible as mobile for NMR, while forming amyloids than in the formation of oligomers. Some of the strong mobile hydration water–protein bonds are disrupted, allowing the protein to form intermolecular connections in the oligomers and amyloids. These new bonds are the ones that hold the oligomers and amyloids together. In the ~5.8 kJ mol^−1^ to 6.01 kJ mol^−1^ potential barrier region, A53T αS amyloids lose the most mobile hydration water, Δ*naa* = 9.8. While the WT αS amyloids lose less mobile hydration water, Δ*naa* = 7.5. The oligomers make approximately half as much water–protein bonds in their formation, namely Δ*naa* = 4.9 in the A53T variants and it is Δ*naa* = 4.0 in the WT αS oligomers. This suggests a more compact form of A53T oligomers, partially explaining why this mutant forms amyloids at a much faster pace than the wild type αS.

The (amyloid–oligomer) mobile hydration difference shows that oligomers are more hydrated than amyloids (Figure 3). As mentioned earlier, this is directly connected to the differences in the compactness of the two protein structures. The difference varies between *naa* = −0.9 and *naa* = −1.8, except at 5.85 kJ mol^−1^ ≤ *E*_a_ ≤ 6.01 kJ mol^−1^, where it extends to as large values as *naa* ≈ −13. The difference changes more for WT than for A53T αS, the average is −1.40(5) with 0.08 variance for WT αS in the 4.96 kJ mol^−1^ ≤ *E*_a_ ≤ 5.85 kJ mol^−1^ potential barrier range while the average is −1.27(4) with 0.04 variance for A53T αS in the same energy range. This means an overall more homogeneous solvent accessible surface of A53T αS. It can be the result of the amyloids having smaller solvent accessible surface per monomer units compared to oligomers.

### 3.2. The Stabilin CTD-Tβ_4_ System

The number of the intermolecular bonds connecting Tβ_4_ and stabilin CTD was determined, to build 1:1 protein complex. The mobile hydration values per amino acid residues were simply added together for the constituting proteins at each potential barrier values, i.e., the *MD*s of these proteins were numerically added. The sum was compared to the measured *MD* of the 1:1 protein complex (Figure 4). The comparison revealed that the sum is greater than the measured values of the 1:1 complex between 5.40(3) kJ mol^−1^ ≤ *E*_a_ ≤ 5.70(5) kJ mol^−1^ potential barrier values. Outside of this range, the sum and measured *MD*s coincide. The difference equals to *naa* = 0.21(1) or 20(1) H_2_O/complex and gives the number of mobile hydration water to protein bonds broken by the formation of the 1:1 protein complex. These broken bonds mean interaction sites that then make new intramolecular bonds, which hold the 1:1 protein complex together. The energy of the bonds falls in the potential-barrier interval of the broken bonds.

Given the sizes of the two proteins, the number of intermolecular bonds is low. Since both of the partners are disordered in isolation, this few intramolecular bonds are not capable of restricting the movement of the two proteins, resulting in a fuzzy complex. In this setup, both Tβ_4_ and stabilin CTD retain most of their structural disorder even in the bound state. Fuzzy complexes, where some level of flexibility is observed after binding, are frequently found in nature [47]. In most cases, when an IDP makes connections with a globular protein, fuzziness is limited to a certain level, but binding of two IDPs can lead to the formation of complexes with extreme fuzziness—where the partners remain mostly disordered [5]. These types of interactions pose a challenge for structural characterization, as most methods are unable to pick up the subtle changes that occur upon binding in such a way. Our results clearly show that Tβ_4_ and stabilin CTD are capable of interacting with each other and remain almost completely disordered in their complex.

## 4. Discussion

The A53T mutants have surplus β-sheets to the wild type α-synucleins according to FTIR, ss-NMR and single-molecule force spectroscopy experiments [21,23,24]. In our results, the additional β-sheet content is indicated by the extra low-potential-barrier constant or almost constant *MD* difference section (Figure 2). The bonds between the monomers in oligomers or amyloids and the secondary structures specific to the aggregates appear as coinciding sections of WT and A53T αSs (thick black lines in the insets of Figure 2). The most strongly bound water molecules start to move close to the melting point of bulk water. Oligomers lose more mobile hydration water than the amyloids as it can be seen on the (amyloid–oligomer) hydration difference. On place of the lost protein–water bonds, new intermolecular connections form during aggregation, which bond the monomers into oligomers or amyloids. The A53T oligomers are more compact than the WT ones according to the (amyloid–oligomer) *MD* differences (Figure 3). This compactness partially gives reason the mutant to form amyloids faster. The oligomers are more hydrated as a result. The A53T αS has more homogeneous solvent accessible surface per monomer units compared to WT αS because, e.g., the mobile hydration in the (amyloid–oligomer) *MD* differences are smaller with half as big variance. The (amyloid–oligomer) *MD* differences being negative, reflect the amyloids having smaller solvent accessible surface per monomer units compared to oligomers.

The sum of the *MD*s of the proteins Tβ_4_ and stabilin CTD is greater than the measured *MD* of the 1:1 protein complex by *naa* = 0.21(1) or 20(1) H_2_O/complex (Figure 4). The Tβ_4_ and stabilin CTD interact with each other while remain almost completely disordered in their complex. Binding of two IDPs like Tβ_4_ and stabilin CTD leads to the formation of a complex with extreme fuzziness [5], the partner proteins remain disordered in the complex. Both constituting proteins in the 1:1 complex retain most of their structural disorder. Tβ_4_ and stabilin CTD connect to each other with only a small number of intermolecular bonds. The intermolecular bonds are not enough numerous to limit the movement of the two proteins in the complex.

## 5. Conclusions

Mobile hydration differences (monomer *MD* subtracted from oligomer and amyloid *MD*s) were calculated for αS variants of different polymerization degrees. The derivative ratios of the *MD*s reflect very sensitively the changes in the trends of the melting curves, and they have characteristically different shapes for the WT and the A53T variants. The constant low *E*_a_ sections in A53T αS difference curves for oligomers and amyloids are caused by the excess β-sheet contents of the mutant compared to WT. The 5.41 kJ mol^−1^ ≤ *E*_a_ ≤ 5.77 kJ mol^−1^ difference *MD* regions belong to interactions linking the monomers into oligomers or amyloids and to the bonds of secondary structures forming in the process of polymerization. The monomers lose much more mobile hydration water while forming amyloids than in the formation of oligomers, reflecting fundamental structural differences between oligomers and amyloids. A part of the strong mobile hydration water–protein bonds are lost and these bonding sites of the protein form intermolecular bonds in the oligomers and amyloids. The new bonds being created connect the constituting proteins into oligomers or amyloids. The amyloid–oligomer difference *MD* showed that there is an overall more homogeneous solvent accessible surface of A53T αS amyloids than the WT variants, suggesting a more densely packed amyloid structure.

We gained information on the intermolecular bonds constructing the αS oligomer and amyloid aggregates from the monomers.

The number of the intermolecular bonds was determined, which connect Tβ_4_ and stabilin CTD in the thymosin-β_4_–stabilin-2 CTD complex. It was found that there are *naa* = 0.21(1) or 20(1) H_2_O/complex such bonds and their bonding energy falls into the 5.40 kJ mol^−1^ ≤ *E*_a_ ≤ 5.70 kJ mol^−1^ range. These results were gained from the comparison of the nominal sum of the *MD*s of the constituting proteins to the measured *MD* of the thymosin-β_4_–stabilin-2 CTD complex and provide further insight into the molecular background of their fuzzy complexes.

## Figures and Tables

**Figure 1 biomolecules-11-00757-f001:**
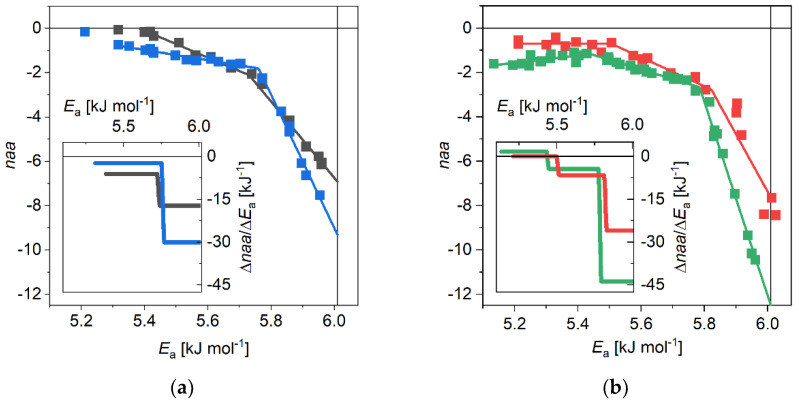
(**a**) Wild type α-synuclein, oligomer-monomer (grey) and amyloid–monomer (blue) melting diagrams; (**b**) A53T α-synuclein, oligomer–monomer (red) and amyloid–monomer (green) melting diagrams. Mol H_2_O per mol amino acid residue (*naa*) vs. potential barrier. Lines are guides to the eye. The insets are the derivative forms of the difference melting diagrams. Mol H_2_O per mol amino acid residue (*naa*) vs. potential barrier. Lines are guides to the eye. The insets are the derivative forms of the difference melting diagrams.

**Figure 2 biomolecules-11-00757-f002:**
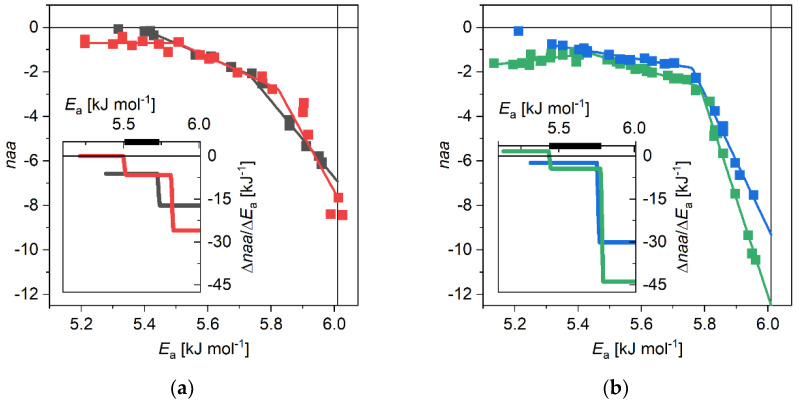
(**a**) Oligomer-monomer melting diagrams for wild type (grey) and A53T (red) α-synuclein; (**b**) Amyloid-monomer melting diagrams for wild type (blue) and A53T (green) α-synuclein. Mol H_2_O per mol amino acid residue (*naa*) vs. potential barrier. Lines are guides to the eye. The insets are the derivative forms of the difference melting diagrams. The thick black lines denote the common sections on the derivative melting diagrams.

**Figure 3 biomolecules-11-00757-f003:**
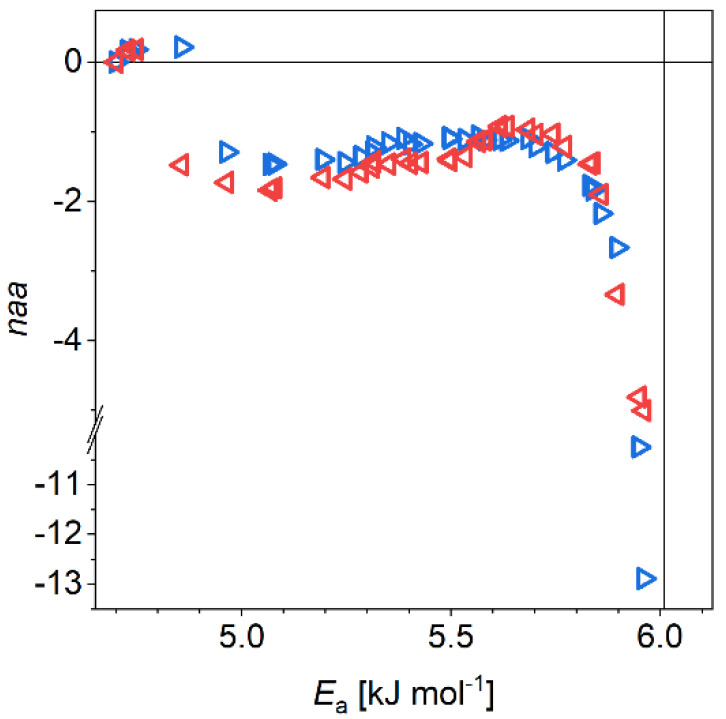
Wild type (red triangles) and A53T (blue triangles) α-synuclein amyloid-oligomer difference melting diagrams. Mol H_2_O per mol amino acid residue (*naa*) vs. potential barrier.

**Figure 4 biomolecules-11-00757-f004:**
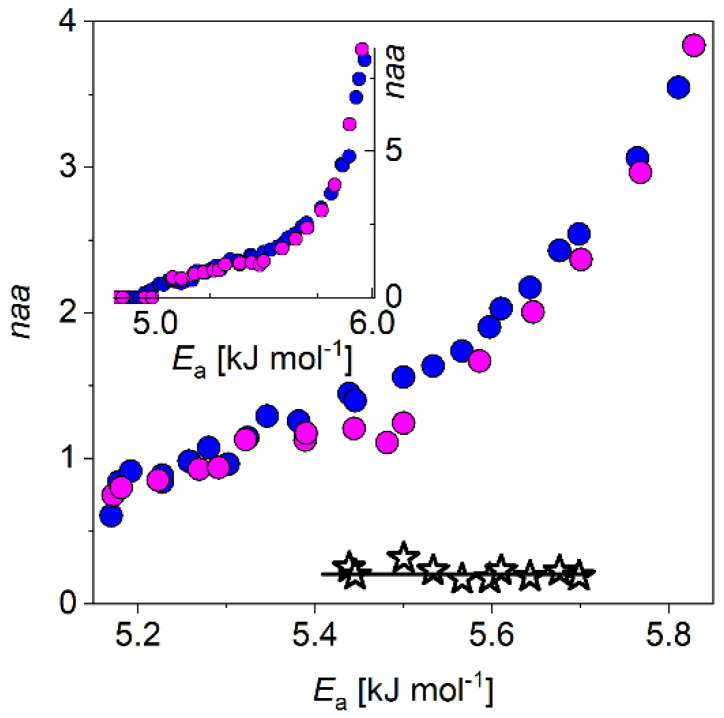
Sum of the measured melting diagrams of isolated thymosin-β_4_ and stabilin-2 CTD (blue circles). Measured melting diagram of thymosin-β_4-_-stabilin-2 CTD complex (magenta circles). Their difference is given by black stars and line. Inset: whole potential barrier range from the no mobile hydration water point to the melting point of water.

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
