# Peer review of "Protein–Protein Connections—Oligomer, Amyloid and Protein Complex—By Wide Line 1H NMR"

_biomolecules, 2021, doi:10.3390/biom11050757_

Round 1

Reviewer 1 Report

Bokor and Tantos measured mobile hydration differences to calculate alpha-S variants of different polymerization degrees and found different shapes for the WT and the A53T variants. They obtained some information on the intermolecular bonds constructing the alpha-S oligomer and amyloid aggregates from the monomers. The information is useful only if some structure-based data would also be provided. I suggest the authors provided more detailed information on the interactions.

Author Response

Structure-based data and more detailed information on the interactions are provided in Refs. [2-5] (new numbering).

Reviewer 2 Report

It is not clear what the purpose if this experimental study is. The selection of systems (protein and protein complexes) under study is not justified and the systems under study do not have much in common. There is no description of the materials (the source and purity of different protein systems; buffer preparations, molarity, incubation time; pH, ionic concentration, temperature ...).  I am not an expert on NMR and I do not know what a melting diagram is. I've never heard of the NMR method used in this paper. Has it ever been validated on a known protein system? It is unclear what the authors identify as "oligomer" or "amyloid". This needs to be clarified for each system under study. How were oligomer or amyloid samples prepared? Oligomers of different sizes coexist with monomers;  if oligomers were separated from monomers, how was this done? What was the size of oligomers? How was the secondary structure of samples under study determined?

Author Response

The purpose of this experimental study is to get known the amount and quality of bonds between constituting parts of a protein aggregate in wild type (WT) and A53T a-synuclein (aS) oligomers, amyloids and in the complex of thymosin-b4–cytoplasmic domain of stabilin-2. The selection of the studied proteins represents a system in which the constituents are different in their degree of polymerization and another system where the constituents interact with each other. The description of the materials can be found in refs. [2,4] (new numbering). The NMR method was validated on several protein systems, see e.g. Refs. K. Tompa et al. Interfacial water at protein surfaces: wide-line NMR and DSC characterization of hydration in ubiquitin solutions. Biophys. J. 2009, 96, 2789-2798; K. Tompa et al. The melting diagram of protein solutions and its thermodynamic interpretation. Int. J. Mol. Sci. 2018, 19, 3571. etc. The terms oligomer and amyloid are used as in the general wide-spread alpha-synuclein literature, therefore more explanation is not needed. All experimental procedures are explained in detail in the updated literature. The extents of the secondary structures were determined by wide-line NMR.

Reviewer 3 Report

The manuscript by Bokor et al. “Protein-protein connections - oligomer, amyloid and protein complex - by wide line 1H NMR” describes two IDP interaction systems, α-synuclein wild type (αSyn) and its mutant A53T and Tb4 and stabilin CTD, to gain information about the bonds holding the protein associations.  The authors employ wide line 1H NMR as appealing technique to tackle several problematic aspects of IDPs, such as their extreme heterogeneity and sensibility to environmental conditions.

The study is well performed and nicely presented.

However, the manuscript has some major and minor issues and should be revised.

• Major issues

1) I’d like to know if the authors use an N-terminal His-tagged version of αSyn or TAT-αSyn. Indeed, even small changes in the first 10 amino acid residues of the αSyn strongly impact its conformation and its interaction with membranes (see for example Bartels, PLoS One 2014; Dikiy, JBC 2014; Maltsev, JACS, 2013 and Maltsev, Biochemistry 2012). Thus, the protein construct used in this study should be entered in Material and Methods section.

2) I am curious to know if, in addition to the formation of oligomers and fibrils, the study performed by wide line 1H NMR allows to discern the presence of aggregates. This point should be explored and discussed.

3) The authors report the presence of an excess of A53T β sheet content compared to the αSyn (page 3 lines 123, 124 and 128). It would be advisable to estimate the percentage of the population folded for the two proteins.

• Minor issues

1) The symbol “β“ is missing from the abstract in line 12

2) Letters (a) and (b) are missing in figure 1.

Author Response

Major issues

1) The a-synuclein proteins were expressed without any tags and Stabilin-2 CTD was expressed using an N-terminal His-tag. We updated the Materials and Methods section to contain this information (lines 96-99).

2) The wide-line 1H NMR experiments clearly show the appearance of aggregates.

3) The quantity of the elevated extent of the A53T β-sheet content is not readily accessible, only the number of the hydrating water molecules are known.

Minor issues

1) corrected

2) corrected

Reviewer 4 Report

The article titled "Protein-protein connections – oligomer, amyloid and protein 2complex – by wide line 1H NMR" authored by Drs Bokor and Tantos, analyzed the behavior of an intrinsically disordered protein (alfa-synuclein) in solution by using the technique NMR. The manuscript is well written and only minor grammatical errors and typos were detected (e.i. in line 12). The section material and methods must be improved before acceptance.

Author Response

The section material and methods was amended. We added more information on the purification of the proteins to contain the main expression and purification methods.

Round 2

Reviewer 2 Report

The authors did not adequately addressed comments in the rebuttal or the revised manuscript.

Reviewer 3 Report

The authors have adequately addresses previous concerns in the revised manuscript.